# Date Fruits as Raw Material for Vinegar and Non-Alcoholic Fermented Beverages

**DOI:** 10.3390/foods11131972

**Published:** 2022-07-02

**Authors:** Elsa Cantadori, Marcello Brugnoli, Marina Centola, Erik Uffredi, Andrea Colonello, Maria Gullo

**Affiliations:** 1Department of Life Sciences, University of Modena and Reggio Emilia, 42123 Reggio Emilia, Italy; elsa.cantadori@unimore.it (E.C.); marcello.brugnoli@unimore.it (M.B.); marinacentola94@gmail.com (M.C.); 2Ponti SpA, 28074 Ghemme, Italy; erik.uffredi@ponti.com (E.U.); andrea.colonello@ponti.com (A.C.)

**Keywords:** date palm, acetic acid bacteria, vinegar, non-alcoholic beverages

## Abstract

Currently, foods and beverages with healthy and functional properties, especially those that claim to prevent chronic diseases, are receiving more and more interest. As a result, numerous foods and beverages have been launched onto the market. Among the products with enhanced properties, vinegar and fermented beverages have a high potential for growth. Date palm fruits are a versatile raw material rich in sugars, dietary fibers, minerals, vitamins, and phenolic compounds; thus, they are widely used for food production, including date juice, jelly, butter, and fermented beverages, such as wine and vinegar. Furthermore, their composition makes them suitable for the formulation of functional foods and beverages. Microbial transformations of date juice include alcoholic fermentation for producing wine as an end-product, or as a substrate for acetic fermentation. Lactic fermentation is also documented for transforming date juice and syrup. However, in terms of acetic acid bacteria, little evidence is available on the exploitation of date juice by acetic and gluconic fermentation for producing beverages. This review provides an overview of date fruit’s composition, the related health benefits for human health, vinegar and date-based fermented non-alcoholic beverages obtained by acetic acid bacteria fermentation.

## 1. Introduction

The current trend to produce fermented beverages and vinegar, including from lesser-known fruits and vegetables, fits with consumers’ demand for a healthier and more sustainable diet. Moreover, health-based recommendations include reducing alcohol consumption, calories from added sugars and limiting the consumption of foods that contain refined grains, especially those with high amounts of added sugars and sodium. The Food and Agriculture Organization of the United Nations (FAO) guidelines for correct nutrition recognized the need to obtain adequate nutrients to reduce the overconsumption of energy, therefore lowering the risk of common chronic diseases, such as diabetes, obesity, cardiovascular disease, and some cancers [1]. Generally, the consumption of foods and beverages with added sugars is discouraged, due to the high calorific content. In addition, the safe use of non-nutritive sweeteners, such as aspartame, is currently under ongoing scientific debate, emphasizing the need to develop natural sweetener products [2]. Due to the above-mentioned trends and health recommendations, non-alcoholic beverages produced without the addition of sugar represent a segment of the food industry that is rapidly growing. The Food and Drug Administration (FDA) considers “non-alcoholic beverages” as all beverages containing less than 0.5% of ethanol by volume [3]. The intake of excessive amounts of ethanol is known to have adverse effects on human health, which include several acute and chronic illnesses [4,5,6]. On the other hand, low alcohol intake is associated with beneficial effects on human health [7,8].

Non-alcoholic beverages or any food (other than alcoholic beverages) are further classified by the FDA into two clusters depending on the final pH. Low-acid beverages have a pH greater than 4.6 and a water activity greater than 0.85; whereas acid beverages have a final pH of 4.6 or below and a water activity greater than 0.85, including beverages such as vinegar, kombucha tea, vinegar beverages, and gluconic beverages [3].

Fruits are optimal substrates for microbial transformations intended to produce non-alcoholic beverages and vinegars, due to their high amount of sugar and fermentative feasibility. Aside from conventional fruits exploited for non-alcoholic beverages and vinegars [9,10,11], there is great interest in the valorization of unconventional ones [12,13,14,15], contributing to open up new opportunities for reducing food loss, and satisfying the consumers’ demand for sustainable and healthy products. 

Date from palm is a versatile raw material, rich in sugar and bioactive compounds, such as phenolic acids, carotenoids, and minerals [16,17]. Due to the peculiar composition, the role of dates and derivatives in reducing the risk of cardiovascular diseases, diabetes mellitus and other illnesses, has been reported [18,19].

However, date production, transformation, and marketing are affected by several issues, such as loss of a high amount of product in field, and no to little available technology for transformation in the production site [20,21]. Previous works highlighted different strategies to valorize date palm fruits and derivative wastes by bioprocesses based on fermentation and enzyme processing. Many added-value products can be obtained such as biopolymers, biofuels, or antibiotics [22,23]. In the food industry, the by-products of dates could be used as source of sucrose substitute for the enzymatic synthesis of fructo-oligosaccharides (FOS), or as raw material to produce high-fructose syrups [24].

Microorganisms that are able to grow and convert date juice into fermented beverages mainly include lactic acid bacteria (LAB), yeasts and acetic acid bacteria (AAB). With regard to microbial bioprocessing, some studies have shown the possibility of using LAB to obtain beverages and probiotics from date derivatives. For instance, date syrup has been previously evaluated as a substrate for producing a probiotic beverage by lactic fermentation [25,26] and for probiotics using date powder, as a low-cost carbon source [27]. Moreover, studies have assessed the possibility to use date by-products to produce lactic acid via lactic fermentation and gluconic acid without the addition of sugars [21,28]. 

The alcoholic fermentation of date juice produces wine, as an alcoholic beverage, or as an intermediate product for further fermentations, operated by AAB. Previous studies state the role of *Saccharomyces cerevisiae* strains as the main yeasts in dates for wine production [29,30,31,32]. Further fermentation of date wine into vinegar is also documented [33,34,35].

Yeasts and AAB are cooperative microorganisms which, through selective fermentations, can promote the valorization of date fruits by alcoholic fermentation followed by acetic fermentation for producing vinegar and low acetic beverages. 

Although AAB, in a single fermentation step, can provide gluconic acid from glucose oxidation. This latter metabolic pathway can be exploited for producing new non-alcoholic beverages from dates.

Although vinegar produced from dates is an existing product, the rational exploitation of AAB for producing both vinegars and low fermented beverages appears underdeveloped. The versatile oxidative metabolism of AAB, and the peculiar traits of species and strains in producing organic acids, mainly acetic and gluconic acid, offer different opportunities to valorize date juices (Figure 1). The production of both acetic and gluconic acid is a common feature of AAB belonging to *Acetobacter*, *Gluconacetobacter,* and *Komagataeibacter;* whereas the production of gluconic acid as primary metabolite is a peculiar trait of *Gluconobacter* members, especially those of *G. oxydans* species [36]. 

This review provides an overview of date palm fruit composition, and discusses fermentation strategies, aimed at producing vinegar and low alcoholic fermented beverages. It presents an analysis of existing products and potential new ones, which exploit the oxidative metabolism of AAB.

## 2. Date Palm Characteristics

Date palm belongs to the *Arecaceae* family and traditionally it is recognized as a valuable beneficial plant. The genus *Phoenix* is composed of 14 species, including *P. dactylifera*, and has been cultivated in the Middle East for 6000 years [37], and *P. sylvestris*, widely known as the wild date palm [37]. The latter, in turn, is widespread in Bangladesh and India where it is also known as date sugar palm, and silver date palm [38,39]. Fruits of *P. sylvestris* are considered beneficial and they are used for their medicinal properties against hyperthermia, nervous debility, back pain, stomachache, toothache, headache, and arthritis [38,40].

In the European market date varieties are mainly imported from Africa. However, in Eastern countries, many varieties of *Phoenix dactylifera* with great biotechnological potential, such as *Saher*, and *Khadrawi,* are cultivated [20].

Date fruits from *P. dactylifera* are much appreciated for their flavor, which has been extensively studied [41,42,43]. The aroma and flavor comes from a complex mixture of volatile compounds. From the volatiloma of 135 varieties of dates, 80 volatile compounds, belonging to acids, alcohols, aldehydes, and esters, were described [41]. The most famous variety of *P. dactylifera* is *Majoul*, which was first imported into the USA from Morocco. Instead, *Ajwa* variety dates can be distinguished from other date varieties due to their higher nutritional properties. Ancient cultures used every part of the plant, such as trunk, leaves, and fruits and for this reason, date palm has been called “the tree of the life” [44].

Currently, over 2000 varieties of date palms are cultivated all around the world, mainly in the Middle East, North Africa, parts of Central and South America, Southern Europe, India, and Pakistan. In 2019, the production of dates globally reached 9.21 million tons [45]. Date palm has high levels of productivity and adaptability allowing the cultivation in desertic areas, such as Saharan regions. Moreover, its fruits have high nutritional values. All these advantages make date palm essential for farmers’ agricultural incomes. According to FAOSTAT, Egypt was the largest producer of palm dates in 2019 with 1.64 million tons, followed by Saudi Arabia with 1.54 million tons, Iran with 1.31 million tons, Algeria with 1.14 million tons, Iraq with 639,315 tons, and Pakistan with 564,904 tons [45]. Furthermore, date palm is resistant to adverse climatic conditions, withstanding temperatures from −6 °C to 50 °C and high salinity levels of soil water. The most appropriate areas for the growth of the palm fruit are arid regions with hot and dry climates and limited rainfall [46]. As the trees are tall, microclimates are generated under the top of the date palm, allowing the production of crops and vegetables that are useful for human and animal sustenance [47].

Date palm is marketed as a high-value fruit crop and low-cost food [48]. According to the geographic origin and quality, dates are marketed as fruits and as several derivatives, such as jam, butter, jelly, date syrup, juice, and non-alcoholic fermented beverages [24].

### The Composition of Date Palm Fruit and Juice

Date fruits are considered to be a primary source of nutrition and energy since they are rich in carbohydrates, mainly sugars, fatty acids, amino acids and minerals [49] (Table 1). In addition, they are well known to be an important source of dietary fiber, thus providing health benefits to humans by preventing diseases and increasing gut well-being [50,51].

The maturation stages, from unripe to ripe, are usually described as *kimri* (greenish color, hard texture), *khalal* (yellowish), *rutab* (softer and sweeter), and *tamer* (dark brown color, soft texture, and highest sweetness) [52]. Date fruits are considered ripe at the *tamer* stage, and they appear dry and firm with a brown/dark color. The nutritional composition of date fruits changes according to the growth stages. For instance, the reduction in phenolic compounds could reach 25% through the ripening stage, represented mainly by carotenoids loss [53,54]. On the other hand, sugar content increases, because of moisture loss during ripening [55]. Total sugar content can reach 81.40% *w*/*w*, 83.41% *w*/*w,* and 88.30% *w*/*w*, in *Barthe*, *Khalas,* and *Deglet Nour* varieties, respectively [56,57,58].

Sugars are mainly represented by glucose and fructose, while sucrose is present at lower concentrations [19,50,57,59,60,61]. Glucose and fructose are main carbon substrates for alcoholic, acetic, and gluconic fermentations. However, other reducing sugars, such as mannose and maltose or polysaccharides, such as cellulose and starch, can be found at low concentrations [47].

Concentrations between 2.70 and 20.25 g/100 g dry matter of fiber have been found by Borchani and her co-workers, who tested the fiber extract of 11 date cultivars [57].

In addition, dates can be defined as the richest and most important source of dietary minerals among other common fruits consumed by humans [48]. A 100 g portion of date fruit is enough to provide the 15% of daily recommended minerals [58].

Low levels of sodium and high levels of potassium make date fruits a recommended food for people suffering from hypertension [57]. Furthermore, date fruits contain other minerals, such as iron, calcium, cobalt, copper, magnesium, manganese, phosphorus, and zinc [62].

The vitamins content of dates is reported to be low. Depending on the stage of ripening and production processes, however, they are considered a good source of folate and vitamin C [16,52,58,63].

**Table 1 foods-11-01972-t001:** Composition of date fruits at *tamer* stage. Values are reported as weight/100 g dry matter. Adapted from [16,50,51,55,57,58,59,62,63].

Dates Composition	Lowest Reported	Highest Reported
	Content [g/100 g]	
Carbohydrates	54.90	88.30
Protein	0.46	3.85
Ash	1.45	2.3
Dietary fiber	2.70	20.25
Fat	0.07	0.57
**Amino acids**	**Content [mg/100 g]**	
Alanine	8.00	342.00
Arginine	2.00	261.00
Aspartame	230.00	450.00
Aspartic acid	2.00	467.00
Cysteine	11.00	114.00
Glutamic acid	40.00	631.00
Glycine	4.00	349.00
Histidine	0.1	76.00
Isoleucine	0.2	465.00
Leucine	0.5	264.00
Lysine	3.00	282.00
Methionine	0.2	219.00
Phenylalanine	0.8	173.00
Proline	12.00	369.00
Serine	6.00	238.00
Threonine	1.00	264.00
Tryptophan	49.5	100.00
Tyrosine	1.00	181.00
Valine	0.5	271.00
**Minerals**	**Content [mg/100 g]**	
Potassium	107.40	916.00
Boron	3.30	5.60
Sodium	32.90	131.00
Calcium	9.50	207.00
Magnesium	47.00	215.55
Phosphorus	13.00	63.00
Iron	0.30	32.76
Copper	0.10	2.90
Cobalt	0.41	1.00
Selenium	0.10	0.32
Zinc	0.10	1.80
Manganese	0.21	5.90
**Vitamins**	**Content [mg/100 g]**	
Folic acid	0.004	0.3
Niacin	0.0004	1.61
Riboflavin (B2)	0.06	0.17
Thiamine (B1)	0.05	0.13
Vitamin C	2.4	17.5

Several studies have reported even the presence of numerous bioactive phytochemicals, such as carotenoids, flavonoids, polyphenols, and steroids, in most of the varieties of date fruits [16,17,33,54,62].

Although the maturation stages can negatively affect the number of phenolic compounds and the antioxidant capacity of date fruits, it is estimated that 100 g of dates contains 250–450 mg of total phenolic compounds (Table 2) [20,53]. Compared to grapes, dates show a higher phenolic content than white/green grapes but less than dark purple/red ones. However, dates are a richer source of phenolic compounds compared to other fruits (Table 2).

Most of the total phenolic compounds in date fruits are phenolic acids, carotenoids, polyphenols, and phytosterols [54]. Phytosterols are the least present whereas phenolic acids are the most common by far (Table 3). The latter includes protocatechuic, gallic, caffeic, p-hydroxybenzoic, vanillic, ferulic, syringic, p-coumaric, and o-coumaric acid [68].

The high prevalence of phenolic compounds and vitamin C is correlated to the strong antioxidant activity of date fruits. Indeed, phenolic acids, anthocyanins, and β-carotene exhibit strong antioxidant potential, playing a key role in the therapeutic effect of date fruits [19]. In addition, such antioxidants could help in reducing chronic inflammation, risk of coronary disease, and development of cancer [17,19,54].

Dates contain a very small quantity of proteins. However, they are rich in amino acids containing essential (e.g., lysine and leucine) and non-essential (e.g., glycine, aspartic acid, and glutamic acid) ones [16,69]. It has also been reported that the concentrated juice, such as date syrup, has strong antioxidant activity, accordingly to the high total phenolic content. This evidence highlights the potential to produce high-value products, especially date syrup, from dates with high polyphenols content and high antioxidant potential [70].

## 3. Fermentation of Date Palm Juice

The feasibility of efficient alcoholic fermentation of date juice is well documented by several studies in which it was intended as single fermentation for obtaining date palm wine and, as the first biological step for obtaining acetic products, such as vinegars.

Palm wine is a generic name for a group of alcoholic beverages obtained from different species of palm, such as *Elaeis guineensis*, *Raphia hookeri*, *Phoenix dactylifera*, *Borassus aethiopum,* and *Cocos nucifera*. Palm wine is the most popular beverage in Africa, and it is claimed that 10 million people in Western Africa consumes it [71]. In Nigeria, palm wine is used during local and traditional events [29].

*Saccharomyces cerevisiae* has been reported as the principal yeast species in traditional date wines, or the most used in date wine production [29,30,31,32]. By using date juice as raw material, wine containing 12% *v*/*v* ethanol by inoculating *S. cerevisiae* var. *ellipsoideus* was produced [46]. Authors reported low acidity levels (0.35–0.54% *v*/*v*) and a pH ranging between 4.0 and 4.2. Similar results, in terms of acidity and pH, were obtained by Awe and co-workers [72], but with a lower ethanol content (9.2% *v*/*v*). Moreover, date wine was richer in vitamin C and protein, compared to a commercial white grape wine [72]. Date extracts were also reported to be suitable for alcoholic and subsequent acetic fermentation for producing vinegar, using a *S. cerevisiae* strain (68 g/L ethanol produced), and a strain of the *Acetobacer aceti* species (45 g/L of acetic acid produced) [73].

In date wine, a decrease in phenolic compounds (e.g. phenolic acids) in juices as a result of the microbial activity is reported. However, date wine is recognized a source of bioactive compounds [74]. In particular, the radical scavenging activity of phenolic compounds present in date wine is associated with a beneficial effect on human health by reducing the risk of coronary disease. Moreover, date wine contains proanthocyanidins, which carry out several advantageous effects on humans such as anticancer, antioxidant, and antidiabetic activity [75,76]. Proanthocyanidins also provides flavor and astringency to beverages [75].

### Vinegar

Vinegar is the result of the activity of yeasts and AAB, as the main microbial groups. Once ethanol is obtained by alcoholic fermentation, AAB converts it into acetic acid by an oxidative fermentation. Primarily, partial oxidations of suitable carbon sources are carried out by the activity of membrane-bound dehydrogenases, located in the periplasmic space of the cell membrane. In terms of the production of acetic acid, the aldehyde dehydrogenase (ALDH; EC 1.2.1.-) and alcohol dehydrogenase (ADH; EC 1.1.2.8) are responsible for the conversion of ethanol into acetaldehyde, and then into acetic acid, respectively [36]. The efficiency of this microbial transformation depends on several factors, such as the raw material composition, the microbial strains, the fermentation regime, and the process parameters [11].

Over time, consumers’ conception of vinegar has evolved from a simple ingredient to a condiment, as part of a more sophisticated choice of consumption. Moreover, along with the growth of public interest in vinegar and healthy beverages, as well as the fact that popularity of vinegar drinks that contain phytochemicals found in the given fruit has increased in recent years [77]. Some of these benefits include enhanced immunity, reduction in risk factors for cardiovascular diseases, improved digestion, appetite suppression, reduced fasting blood glucose, reduced blood pressure, and serum cholesterol [78].

The pharmacological potentials of date palm fermented products are also documented. Different studies have shown the anti-hyperlipidemic, anti-obesity, antioxidant, and immune-stimulating activities of date vinegar made from date flesh and pits (seeds) [79,80,81,82]. Moreover, homemade date vinegar produced from date waste is a traditional product in Iran, widely consumed for its antimicrobial properties [83].

Matloob [35] tested the production of vinegar utilizing *Khistawi* date juice as a raw material. The alcoholic fermentation was carried out by a bakery yeast (*S. cerevisiae*), while the aerobic oxidation of ethanol to acetic acid by an *Acetobacter* strain. The pH of produced vinegars ranged between 2.40 and 3.26 with a minimum acidity of 4.00%, which is the minimum value required by the North American legislations [84].

The effect of fermentation on phenolic content of vinegar produced from different dates’ cultivars was evaluated by Matloob and Balakit, who observed a decrease during alcoholic fermentation from an initial value of 1211.8 mg GAE/L to 1179.8 mg GAE/L (−2.6%). However, the subsequent acetic fermentation provided a significant increase (+20.1%) of total phenolic content in *Khistawi* date vinegar, reaching a peak value of 1453.4 mg GAE/L during 14 days of fermentation [33]. The increase in phenolic compounds in vinegar brewing, observed by several authors, could be due to two different reasons [85,86,87]. First, the acids produced during acetic fermentation could degrade the glucoside bonds of the phenolic compounds, leading to the liberation of compounds with different structure [33,88]. The second reason could be the enzymatic conversion of high molecular weight polyphenol compounds into small molecules having higher biological activity [89,90,91].

The high content of phenolic compounds confers to date vinegar inhibitory effects against oxidative reactions [80,92,93,94,95]. Ali and co-workers [80] tested daily consumption effects of date vinegars on hypercholesterolemic adults. Outcomes showed a positive correlation between date vinegar consumption and the reduction in serum total cholesterol, low-density lipoprotein, and apolipoprotein B (Apo B) concentration. Furthermore, an increase in high-density lipoprotein (HDL) concentration was observed. Health benefits were mainly related to acetic acid, dietary fiber, and phenol compounds concentration. Indeed, acetic acid lowers the levels of the substrate required for serum cholesterol production by suppressing the sterol regulatory element, binding protein mRNA levels. The combination of HDL higher levels and acetic acid effects could lower cardiovascular diseases risk by decreasing Apo B concentration.

Beneficial effects of low acidity date vinegar/beverage supplemented with garlic juice was also stated as a means of lowering the total cholesterol content [79]. Researchers hypothesized that the high intake of bioactive compounds such as carotenoids, fiber and potassium caused an obliteration of intestinal lipid saturation. This evidence highlights the potential to produce functional beverages based on date juice with the added value of benefits for hyperlipidaemic adults.

Date vinegar also contains unsaturated fatty acids, such as oleic, palmitoleic, linoleic, and linolenic acids even though the highest amount is available in date seeds [55].

## 4. Exploring Innovative Date Fruits Products by Acetic Acid Bacteria Fermentation

The interest of consumers in vinegars and low acetic acid content beverages, drives the industry to explore unconventional raw materials as having the potential to enhance functional properties of the end-products. Approaches include process improvements by innovative treatment of the raw material to reduce the loss of bioactive compounds. For instance, Siddeeg and co-workers [34] evaluated the influence of ultrasound (US), pulsed electric field (PEF), and the combination of both techniques on the quality of date vinegar compared to untreated ones. Prior to alcoholic fermentation, dates were treated with PEF, US, or PEF + US. Outcomes showed no changes or minor changes in color parameters, pH, total titratable acidity, and residual ethanol concentration. Moreover, PEF + US samples presented a notable increase in total phenolic and flavonoid content, free amino acid content, and volatile components. Date vinegars treated with PEF + US, PEF or US obtained a higher overall acceptability score compared to the untreated samples.

A great opportunity to valorize date juice arises from selected AAB fermentation which, based on the know-how acquired by vinegar production, can drive the development of new processes and products (Figure 1). An optimization strategy, by using full factorial design approach, has been applied for studying traditional date vinegar in Algeria [96]. On the basis of current knowledge, both conventional processes for obtaining vinegar can be modulated to obtain low acetic acid beverages with enhanced properties. Moreover, there is high potential of innovation in terms of gluconic acid fermentation. Contrary to vinegar production, non-alcoholic beverages containing gluconic acid are produced in one fermentation step process, which consist in the oxidation of glucose to gluconic acid. Glucose oxidation leads to the production of glucono-δ-lactone acid in a reaction catalyzed by the membrane-bound pyrrolo-quinolinequinone-dependent gluconate dehydrogenase (GDH; EC. 1.1.5.2). Glucono-δ-lactone is stable in acid conditions, but it can spontaneously hydrolyze to gluconic acid under neutral and alkaline conditions, or can be converted to gluconic acid by a membrane-bound gluconolactonase (GNL; EC:3.1.117) [36]. Even though some AAB are able to produce a high amount of gluconic acid, this metabolic pathway is not fully exploited for producing acid beverages, yet. Fermented beverages containing gluconic acid can take advantage of the “mild” properties of gluconic acid, which provide appreciated sensorial properties and, at the same time, preservative properties [97].

As reported in this review, over the past few years, low alcoholic and non-alcoholic beverages with functional properties are emerging products with an outstanding potential. Kombucha tea is an example of a non-alcoholic fermented beverage consumed for its beneficial effects on human health [98,99,100]. Kombucha tea is produced by fermenting sweetened tea with a microbial community composed of yeasts and AAB, of which most of the strains belong to the genus *Komagataeibacter* [101,102]. Recently, new Kombucha tea-based beverages have appeared on the market. These new products include a number of beverages produced by the basic practice to obtain the original Kombucha tea, but they can be flavored or obtained using raw materials different from tea or tea blended with suitable fermentable juices. Khosravi and co-workers [103] tested the date syrup as a substrate for producing a Kombucha-like beverage. Date syrups with different °Brix values were inoculated with a Kombucha starter culture and final products were compared to a standard Kombucha tea. Outcomes revealed faster changes of fermentation parameters, such as pH and titratable acidity in date syrup Kombucha. During Kombucha tea production, AAB oxidize ethanol to produce acetic acid, whereas glucose and fructose are used to produce gluconic acid, glucuronic acid, and bacterial cellulose [36,101,103,104]. Therefore, Khosravi and co-workers’ study [103] proved that date syrup is a good raw material for Kombucha-like beverages production stating dates’ product as a potential substrate for vinegar beverages and gluconic beverages too.

## 5. Conclusions

Among the strategies to valorize fruits and vegetables, microbial fermentation is one of the most valuable tools, helping to provide several sustainable and healthy foods and beverages.

Date fruits and their derivatives are versatile raw materials for microbial fermentation, being rich in fermentable sugars and bioactive compounds. Although they are already on the market as fruits and transformed products, a considerable number of dates are lost annually. On the other hand, efficient microbial transformations could open up the opportunity to satisfy consumers’ demand of healthy and sustainable foods, while reducing food loss. When considering the AAB role for valorization of date fruits, both acetic acid and gluconic acid fermentations can be exploited for obtaining vinegars, low acetic and gluconic beverages. Although vinegars from dates are already produced, little evidence is available on low acetic and gluconic beverages, which could be a valuable industrial segment considering the interest in non-alcoholic, non-sugared and functional beverages. The know-how acquired in the vinegar field could drive the rational design of processes aimed at exploiting AAB metabolic potential in performing both acetic and gluconic oxidation.

## Figures and Tables

**Figure 1 foods-11-01972-f001:**
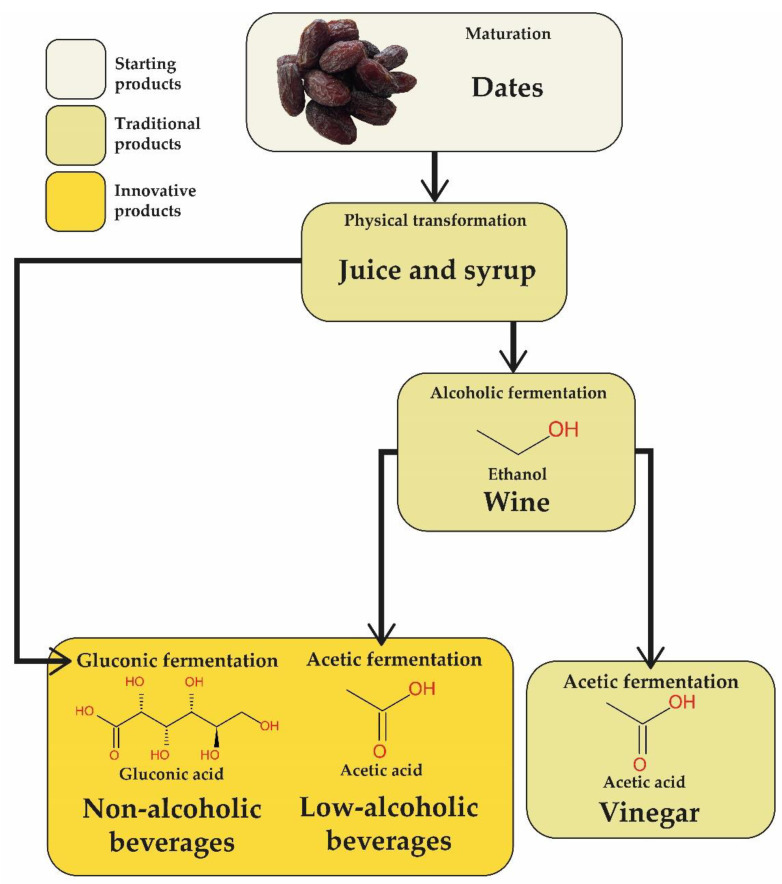
Traditional and innovative date fruit products. Box’s color indicates the type of products. Chemical compounds and their structure indicate the specific type of beverage.

**Table 2 foods-11-01972-t002:** Total phenolic contents of various fruits estimated by Folin-Ciocalteu method. Adapted from [20,33,53,64,65,66,67].

Fruits	Phenolic Content (mg GAE/100 g Fresh Weight)
Date	326
Green grape	201
Dark purple grape	397
Kiwifruit	112
Orange	243
Plum	311
Apple	100
Pear	125
Raspberry	267

**Table 3 foods-11-01972-t003:** Bioactive compounds in date fruit at *tamer* stage (mg/100 g fresh weight). Adapted from [20].

Bioactive Components	Content [mg/100 g]	
	Lowest Reported	Highest Reported
Phenolic acids	20.24	64.44
Carotenoids	0.03	2.90
Anthocyanins	0.24	1.52

## Data Availability

Not applicable.

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
