# Peer review of "Date Fruits as Raw Material for Vinegar and Non-Alcoholic Fermented Beverages"

_foods, 2022, doi:10.3390/foods11131972_

Round 1
Reviewer 1 Report
Dear Authors,
I have carefully read the manuscript and find it very insightful and significant for the scientific community.
However, certain issues must be answered. In my view, the manuscript needs to be a major revision.
My suggestions are listed as follows:
- the microbiological aspect of the topic is missing for obtaining the complete review; I need to require the addition of this aspect in the paper.
- So many paragraphs in the paper, please try to connect them mutually
- change the font in the table in order to uniform it with text.
Author Response
Reviewer 1
Dear Authors,
I have carefully read the manuscript and find it very insightful and significant for the scientific community.
However, certain issues must be answered. In my view, the manuscript needs to be a major revision.
My suggestions are listed as follows:
- the microbiological aspect of the topic is missing for obtaining the complete review; I need to require the addition of this aspect in the paper.
Thank you very much for your suggestions. We revised the manuscript focusing more in depth on the microbiological aspects. Revisions were done in different part of the manuscript from the Abstract to Conclusions.
Abstract
Line 13-20: “Date palm fruits are a versatile raw material, rich in sugars, dietary fibers, minerals, vitamins, and phenolic compounds, thus they are widely used for food production, including date juice, jelly, butter, and fermented beverages, such as wine and vinegar. Moreover, their composition makes them suitable for the formulation of functional foods and beverages. Microbial transformations of date juice include alcoholic fermentation for producing wine as end product or as a substrate for acetic fermentation. Lactic fermentation is also documented for transforming dates juice and syrup. However, considering acetic acid bacteria, little evidence is available on the exploitation of date juice by gluconic fermentation for producing beverages.”
Introduction:
Lines 69-70: “Microorganisms able to grow and convert date juice into fermented beverages mainly include lactic acid bacteria (LAB), yeasts and acetic acid bacteria (AAB).”
Lines 77-80: “By alcoholic fermentation of date juice wine is produced, as an alcoholic beverage or as an intermediate product for further fermentations operated by AAB. Previous studies report the role of Saccharomyces cerevisiae strains as main yeasts in date for wine production [29–32]. Further fermentation of date wine into vinegar is also documented [33–35].”
Lines 95-98: “In this review starting from an overview on date palm composition, fermentation strategies aimed at producing vinegar and low alcoholic fermented beverages from dates are discussed. An analysis of existing products and potential new ones, exploiting the oxidative metabolism of AAB, are presented.”
- So many paragraphs in the paper, please try to connect them mutually
According to your suggestions to provide a more connected text, the structure of the manuscript was modified, as follow:
The paragraph “The composition of date palm fruit and juice” is now a subparagraph of “2. Date palm characteristics”, named “2.1 The composition of date palm fruit and juice”.
The paragraph “Vinegar” is now a subparagraph of “3. Fermentation of date palm juice”, named “3.1 Vinegar”.
Line 142: “2.1 The composition of date palm fruit and juice”
Line 235: “3.1 Vinegar”
- change the font in the table in order to uniform it with text.
The correction has been done.
Reviewer 2 Report
Valorization of date fruits: from juice to vinegar and non-alcoholic fermented beverages
Comments:
The main results of this review is missing in the abstract.
Line no 9: What type of functional properties? Give Example
Line no 30: Food with sugar will always low in nutritional content? Justify
Line no 37: What will be effect of ethanol percentage on final drink?
Line no 46: What’s the reason behind to prefer unconventional ones?
Line no 49: Mention bioactive compounds.
Line no 52. Please check this paper and add one sentence about the health benefits on date palm and its impact on various diseases, here are the papers:
Functional Food and Nutra-pharmaceutical Perspectives of Date (Phoenix dactylifera L.) Fruit. Journal of Food Biochemistry, 44: e13332.
Therapeutic potential of date palm against human infertility: A review. Metabolites, 11(6), 408).
Line no 54. Please add the citation for this statement. Check this paper (Efficient utilization of date palm waste for the bioethanol production through Saccharomyces cerevisiae strain. Food Science and Nutrition, 9(4), 2066–2074.)
Line no 56: Does substitution will produce same effects?
Line no 68: AAB? Use full form first and then abbreviate
Line no 72: introduction needs improvement, try to write in linked manner in order to understand easily.
Line no 96: Graphical representation is appreciable but similar pattern for font should be used.
Line no106: Briefly discuss flavor profile, the main compound of flavor.
Line no 134: Maturation stages with nutritional impact? Discuss
Line no 146: Cultivars? Best cultivar according to nutritional content. Discuss
Line no 149: What is the relation between total sugar and maturation stages?
Line no 157: Give reference
Line no 184: Phenolic compound? Discuss the main ones
Line no 206: Awe? First give full form.
Line no 212: In the whole manuscript there is no discussion of phenolic compounds? Give example of main phenolic content.
Line no 228: Give references?
Line no 313: Conclusion is too short to give the whole idea of manuscript.
Author Response
Reviewer 2
Valorization of date fruits: from juice to vinegar and non-alcoholic fermented beverages
Comments:
The main results of this review is missing in the abstract.
Thank you for the comment. The Abstract was revised, according to your suggestions.
Lines 16-20: “Microbial transformations of date juice include alcoholic fermentation for producing wine as end product or as a substrate for acetic fermentation. Lactic fermentation is also documented for transforming dates juice and syrup. However, considering acetic acid bacteria, little evidence is available on the exploitation of date juice by gluconic fermentation for producing beverages.”
Line no 9: What type of functional properties? Give Example
Thank you for the suggestions. The sentence was modified.
Line 9: “… especially those claimed to prevent chronic diseases,...”
Line no 30: Food with sugar will always low in nutritional content? Justify
Thank you very much for your suggestion. We are sorry for not explaining the concept properly. Foods containing sugar are not poor in nutritional content since a lot of nutrients can be present. However, foods and beverages with sugar added during the production process could contain high amounts of sugar which could have a negative effect on human health. We rearranged part of the paragraph to clarify the concept that the consumption of foods and beverages with added sugar is discouraged.
Line 35-36: “Generally, the consumption of foods and beverages with added sugars is discouraged, due to the high caloric content”.
Line no 37: What will be effect of ethanol percentage on final drink?
Thank you for the suggestion. The sentence has been modified as follow:
Lines: 42-45: “The intake of excessive amounts of ethanol is known to have adverse effects on human health, which include a number of acute and chronic illnesses [4-6]. On the other hand, low alcohol intake is associated to beneficial effect of human health [7,8].”
Line no 46: What’s the reason behind to prefer unconventional ones?
Thank you for the suggestion. The sentence was rewritten as follow:
Line 54-56: “… contributing to open new opportunities for reducing food loss and satisfying the consumers’ demand for sustainable and healthy products.”
Line no 49: Mention bioactive compounds.
Thank you for the suggestion. Main bioactive compounds found in date palm were reported.
Line 58: “such as phenolic acids, carotenoids and minerals [16,17].”
Line no 52. Please check this paper and add one sentence about the health benefits on date palm and its impact on various diseases, here are the papers:
Functional Food and Nutra-pharmaceutical Perspectives of Date (Phoenix dactylifera L.) Fruit. Journal of Food Biochemistry, 44: e13332.
Therapeutic potential of date palm against human infertility: A review. Metabolites, 11(6), 408).
Thank you for suggesting us this additional literature that we now included in the manuscript.
Lines 58-60: “Due to the peculiar composition, the role of dates and derivates in reducing the risk of cardiovascular diseases, diabetes mellitus and other illnesses has been reported [18,19].”
Line no 54. Please add the citation for this statement. Check this paper (Efficient utilization of date palm waste for the bioethanol production through Saccharomyces cerevisiae strain. Food Science and Nutrition, 9(4), 2066–2074.)
Thank you for your suggestion. The following two citations [22,23] were included:
- Ahmad, A.; Naqvi, S.A.; Jaskani, M.J.; Waseem, M.; Ali, E.; Khan, I.A.; Manzoor, M.F.; Siddeeg, A.; Aadil, R.M. Efficient utilization of date palm waste for the bioethanol production through Saccharomyces cerevisiae strain. Food Sci. Nutr. 2021, 9, 2066–2074, doi: 10.1002/fsn3.2175
- Khaman, P.N.; AlMaadeed, M.A. Improvement of ternary recycled polymer blend reinforced with date palm fibre. Mater. Des. 2014, 60, 532–539, doi:10.1016/j.matdes.2014.04.033
Line no 56: Does substitution will produce same effects?
Thank you for the comments. The meaning was to underline the possibility to use dates by-products as a source of sucrose. The sentence has been modified as followed:
Lines 66-68: “In the food industry date’s by-products could be used as source of sucrose substitute for the enzymatic synthesis of fructooligosaccharides (FOS) or as raw material to pro-duce high-fructose syrups [24].”
Line no 68: AAB? Use full form first and then abbreviate
The full name was added in line 70.
Line no 72: introduction needs improvement, try to write in linked manner in order to understand easily.
The Introduction section was rearranged, according to your suggestions.
Line no 96: Graphical representation is appreciable but similar pattern for font should be used.
We are sorry for the mistake. Font has been changed according to text’s pattern.
Line no106: Briefly discuss flavor profile, the main compound of flavor.
Thank you very much for your suggestion. We briefly discussed the flavor profile and main class of volatile compounds found in dates. Three new references have been added (references [41], [42], and [43]) to better explain which are the most dominant classes of volatile compounds.
Line 116-119: “Date fruits from P. dactylifera are very appreciated for their flavor which has been extensively studied [41–43]. The aroma and flavor origin from a complex mixture of volatile compounds. From the volatiloma of 135 varieties of dates, 80 volatile compounds, belonging to acids, alcohols, aldehydes, and esters, were described [41].”
Line no 134: Maturation stages with nutritional impact? Discuss
The sentence was modified as follow:
Lines 150-155: “The nutritional composition of date fruits changes according to the growth stages. For instance, the reduction of phenolic compounds could reach 25% through the ripening stage, represented mainly by carotenoids loss [53,54]. On the other hand, sugar content increase, as a result of moisture loss during ripening [55]. Total sugar content can reach 81.40% w/w, 83.41% w/w and 88.30% w/w, as observed in Barthe, Khalas and Deglet Nour varieties, respectively [56–58].”
Line no 146: Cultivars? Best cultivar according to nutritional content. Discuss
We substituted the term “cultivars” with “varieties” and list 3 varieties with high sugar concentration. Three more refences were added (references [56], [57], and [58]).
Lines 154-155: “Total sugar content can reach 81.40% w/w, 83.41% w/w and 88.30% w/w, as observed in Barthe, Khalas and Deglet Nour varieties, respectively [56–58].”
Line no 149: What is the relation between total sugar and maturation stages?
Thank you for the comment. The sugar amount increases along with the maturation stage.
Lines 153-154: “On the other hand, sugar content increase, as a result of moisture loss during ripening [55].”
Line no 157: Give reference
We added the following reference in line 164: Al-Farsi, M.A.; Lee, C.Y. Nutritional and functional properties of dates: a review. Crit. Rev. Food Sci. Nutr. 2008, 48, 877–887, doi:10.1080/10408390701724264
Line no 184: Phenolic compound? Discuss the main ones
The three main classes of phenolic compounds (phenolic acids, anthocyanins, and β- carotene) have been discussed. In addition, we reported the main therapeutic effects of such antioxidants.
Lines 197-200: “Indeed, phenolic acids, anthocyanins, and β-carotene exhibit strong antioxidant potential playing a key role in the therapeutic effect of date fruits [19]. In addition, such antioxidants could help in reducing chronic inflammation, risk of coronary disease, and development of cancer [17,19,54].”
Line no 206: Awe? First give full form.
Line 222, we added reference [72] corresponding to: Awe, S.; Nnadoze, S. Production and microbiological assesment of date palm (Phoenix dactylifera L.) fruit wine. Br. Microbiol. Res. J. 2015, 8, 480–488, doi:10.9734/bmrj/2015/16867
Line no 212: In the whole manuscript there is no discussion of phenolic compounds? Give example of main phenolic content.
Thank you for the suggestion. We are sorry for the lack in phenolic compounds discussion. In the whole manuscript we enlarged the discussion on phenolic compounds. Then, more details on the phenolic compounds of date wine have been added.
Line no 228: Give references?
The statement is based on literature comparison. An indication about finding the values in Table 2 (along with related references) has been added.
Line 178-183: “Although the maturation stages can negatively affect the number of phenolic compounds and the antioxidant capacity of date fruits, it is estimated that 100g of date contains 250-450 mg of total phenolic compounds (Tab. 2) [20,53]. Compared to grapes, dates show a higher phenolic content than white/green grape but less than dark purple/red one. However, dates are a richer source of phenolic compounds com-pared to other fruits (Tab. 2).”
Line no 313: Conclusion is too short to give the whole idea of manuscript.
The Conclusions section was completely rearranged.
Lines 344-359: “Among strategies to valorize fruits and vegetables, microbial fermentation is one of the most valuable tools, contributing to provide several sustainable and healthy foods and beverages.
Date fruits and their derivates are versatile raw materials for microbial fermentation, being rich in fermentable sugars and bioactive compounds. Although dates are already placed on the market as fruits and transformed products, annually a considerable amount of dates is lost. On the other hand, efficient microbial transformations could open the opportunity to satisfy consumers’ demand of healthy and sustainable foods, while reducing food loss. Considering the AAB role for valorization of date fruits, both acetic acid and gluconic acid fermentations can be exploited for obtaining vinegars, low acetic and gluconic beverages. Although vinegars from dates are already produced, few evidence are available on low acetic and gluconic beverages, which are a valuable industrial segment considering the interest for non alcoholic, not sugared and functional beverages. The know-how acquired in vinegar field could drive the rational design of processes aimed at exploiting AAB metabolic potential in performing both acetic and gluconic oxidation.”

Reviewer 3 Report
The manuscript „ Valorization of date fruits: from juice to vinegar and non-alcoholic fermented beverages“ has important information, followed by explanations of the potential of date fruits throughout valorization as fermented beverages and vinegar The manuscript provides important information on this topic, but the structure of the manuscript as well as its elaboration should be significantly improved. The title of the paper adequately reflects the subject under investigation in the proposed study.
There are issues to be addressed:
Abstract is concise and clearly written, with a good command of English, and clear representation of the aim of the paper. Containing exactly 200 words, it can be said that it does meet the demands of the journal Foods (200 max). Furthermore, it is adequately structured: background of the proposed research, the method used, and main conclusions were mentioned.
Line 15
Insert dot (e.g. date pits)
Introduction
The authors clearly represented the type of food product that is being presented in the manuscript, as well as the legal regulations that prescribe its characteristics. In the final paragraphs of the introductory section the authors explain what the core of their review is. However, the introductory part should be covered with a higher number of references that covers information about the basic chemical composition of dates, the possibility of application in the food industry through bioprocessing as well as the fermentation process in general and as non-alcoholic fermented beverages.
Line 68
Please enter acetic acid bacteria next to the abbreviation AAB. It is not specified.
Line 129
Please specified “The composition of date palm fruit and juice” as subtitle
Line 133
Add more references and comments which evaluated nutrition and proximate analysis of date palm fruit.
Line 144-149
Enter more references and evidence on the sugar composition in date fruit, elaborate its influence on the fermentation process, from the position of primary substrate for microorganisms’ growth and development.
Line 216
Specified Vinegar as subtitle
Line 228-236
The statements in these paragraphs as well as Table 3 refers to the polyphenol composition of date fruits, therefore they should be in the section related on polyphenols in raw date fruit, in the section The composition of date palm fruit and juice.
Line 248
Add more study evidences regarding polyphenols fate during fermentation process and vinegar production.
Line 287-291
Add references related to gluconic acid fermentation. Explain more precisely characteristics of the gluconic fermentation process and the final product.
Conclusions
The conclusions are significantly abbreviated. They need to be improved and supplemented with more conclusions in line with manuscript content. The conclusions contain information about by-products and waste amounts from the date processing industry, although waste issue has not been described in the manuscript.
Author Response
Reviewer 3
The manuscript „ Valorization of date fruits: from juice to vinegar and non-alcoholic fermented beverages“ has important information, followed by explanations of the potential of date fruits throughout valorization as fermented beverages and vinegar The manuscript provides important information on this topic, but the structure of the manuscript as well as its elaboration should be significantly improved. The title of the paper adequately reflects the subject under investigation in the proposed study.
There are issues to be addressed:
Abstract is concise and clearly written, with a good command of English, and clear representation of the aim of the paper. Containing exactly 200 words, it can be said that it does meet the demands of the journal Foods (200 max). Furthermore, it is adequately structured: background of the proposed research, the method used, and main conclusions were mentioned.
Thank you very much for your comment.
Line 15 Insert dot (e.g. date pits)
Dot was inserted.
Introduction
The authors clearly represented the type of food product that is being presented in the manuscript, as well as the legal regulations that prescribe its characteristics. In the final paragraphs of the introductory section the authors explain what the core of their review is. However, the introductory part should be covered with a higher number of references that covers information about the basic chemical composition of dates, the possibility of application in the food industry through bioprocessing as well as the fermentation process in general and as non-alcoholic fermented beverages.
The Introduction section was rearranged, according to your suggestions. The section has been covered with more references along with potential application in the food industry.
Line 68. Please enter acetic acid bacteria next to the abbreviation AAB. It is not specified.
The correction has been done.
Line 129. Please specified “The composition of date palm fruit and juice” as subtitle
The correction has been done.
Line 133. Add more references and comments which evaluated nutrition and proximate analysis of date palm fruit.
According to your suggestion, this part was rearranged.
Lines 150-155: “The nutritional composition of date fruits changes according to the growth stages. For instance, the reduction of phenolic compounds could reach 25% through the ripening stage, represented mainly by carotenoids loss [53,54]. On the other hand, sugar content increase, as a result of moisture loss during ripening [55]. Total sugar content can reach 81.40% w/w, 83.41% w/w and 88.30% w/w, as observed in Barthe, Khalas and Deglet Nour varieties, respectively [56–58].”
Line 144-149. Enter more references and evidence on the sugar composition in date fruit, elaborate its influence on the fermentation process, from the position of primary substrate for microorganisms’ growth and development.
Thank you very much for your suggestion. We updated the references with studies that characterized the composition of numerous varieties of date fruits from different countries across the world. A part focusing on the utilization of fermentable sugars, such as glucose, fructose and sucrose, by acetic acid bacteria has been added. We supported the discussion.
Line 158-165: “Sugars are mainly represented by glucose and fructose, while sucrose is present at low concentrations [19,50,57,59–61]. Glucose and fructose are main carbon substrates for alcoholic, acetic and gluconic fermentations.
We also rearranged “Vinegar” subparagraph adding more explanation on the fermentation and considering the pathway to transform ethanol into acetic acid. The same was done for gluconic acid formation from glucose (paragraph 4. Exploring innovative date fruits products by acetic acid bacteria fermentation).
Lines 237-246: “Vinegar is the result of at least the activity of yeasts and AAB, as main microbial groups. Once ethanol is obtained by alcoholic fermentation, AAB convert it to acetic acid by an oxidative fermentation. Basically, partial oxidations of suitable carbon sources, are carried out by the activity of membrane-bound dehydrogenases, located in the periplasmic space of the cell membrane. Considering the production of acetic acid, the aldehyde dehydrogenase (ALDH; EC 1.2.1.-) and alcohol dehydrogenase (ADH; EC 1.1.2.8) are responsible for the conversion of ethanol into acetaldehyde and then into acetic acid, respectively [36]. The efficiency of this microbial transformation depends on several factors, such as the raw material composition, the microbial strains, the fermentation regime, and the process parameters [11].”
Lines 313-320: “Contrary to vinegar production, non-alcoholic beverages containing gluconic acid are produced in one fermentation step process, which consist in the oxidation of glucose to gluconic acid. Glucose oxidation leads to the production of glucono-d-lactone acid in a reaction catalysed by the membrane-bound pyrrolo-quinolinequinone-dependent gluconate dehydrogenase (GDH; EC. 1.1.5.2). Glucono-d-lactone is stable in acid conditions, but it can spontaneously hydrolyse to gluconic acid under neutral and alkaline conditions or can be converted to gluconic acid by a membrane bound gluconolactonase (GNL; EC:3.1.117) [36]”.
Line 216 Specified Vinegar as subtitle
The correction has been done.
Line 228-236 The statements in these paragraphs as well as Table 3 refers to the polyphenol composition of date fruits, therefore they should be in the section related on polyphenols in raw date fruit, in the section The composition of date palm fruit and juice.
Thank you very much for your suggestion. We rearranged the statement and inserted Glucose and fructose are main carbon substrates for alcoholic, acetic and gluconic fermentations. in the section The composition of date palm fruit and juice. Table 3 has become Table 2.
Line 178-183: “Although the maturation stages can negatively affect the number of phenolic compounds and the antioxidant capacity of date fruits, it is estimated that 100g of date contains 250-450 mg of total phenolic compounds (Tab. 2) [20,53]. Compared to grapes, dates show a higher phenolic content than white/green grape but less than dark purple/red one. However, dates are a richer source of phenolic compounds com-pared to other fruits (Tab. 2).
Table 2: Total phenolic contents of various fruits. Adapted from [20,33,53,64–67]”
Line 248 Add more study evidences regarding polyphenols fate during fermentation process and vinegar production.
Thank you for the suggestion. We specified the fate of phenolic compounds during fermentation process adding references to support the statement.
Line 260-274: “The pH of produced vinegars ranged between 2.40 and 3.26 with a minimum acidity of 4.00%, which is the minimum value demanded by the North American legislations [84].
The effect of fermentation on phenolic content of vinegar produced from different dates’ cultivars was evaluated by Matloob and Balakit, who observed a decrease during alcoholic fermentation from an initial value of 1211.8 mg GAE/L to 1179.8 mg GAE/L (-2.6%). However, the subsequent acetic fermentation provided a significant increase (+20.1%) of total phenolic content in Khistawi date vinegar, reaching a peak value of 1453.4 mg GAE/L during 14 days of fermentation [33]. The increase of phenolic compounds in vinegar brewing, observed by several authors, could be due to two different reasons [85-87]. First, the acids produced during acetic fermentation could degrade the glucoside bonds of the phenolic compounds, leading to the liberation of compounds with different structure [33,88]. The second reason could be the enzymatic conversion of high molecular weight polyphenol compounds into small molecules with higher biological activity during acetic fermentation [89-91].”
Line 287-291 Add references related to gluconic acid fermentation. Explain more precisely characteristics of the gluconic fermentation process and the final product.
Thank you for your suggestion. We explained the reactions that lead to the production of gluconic acid from glucose.
Line 312-320: “Contrary to vinegar production, non alcoholic beverages containing gluconic acid are produced in an one fermentation step process, which consist in the oxidation of glucose to gluconic acid. Precisely, in a reaction catalyzed by the membrane-bound pyrroloquinoline-quinone-dependent gluconate dehydrogenase (GDH; EC. 1.1.5.2), glucose is oxidized into glucono-d-lactone acid, which is converted to gluconic acid by a membrane-bound gluconolactonase (gnl; EC:3.1.1.17) [36]. Even though some AAB are able to produce high amount of gluconic acid, this metabolic pathway is not fully exploited for producing acid beverages, yet.”
Conclusions
The conclusions are significantly abbreviated. They need to be improved and supplemented with more conclusions in line with manuscript content. The conclusions contain information about by-products and waste amounts from the date processing industry, although waste issue has not been described in the manuscript.
The Conclusions section was completely rearranged.
Lines 344-359: “Among strategies to valorize fruits and vegetables, microbial fermentation is one of the most valuable tools, contributing to provide several sustainable and healthy foods and beverages.
Date fruits and their derivates are versatile raw materials for microbial fermentation, being rich in fermentable sugars and bioactive compounds. Although dates are already placed on the market as fruits and transformed products, annually a considerable amount of dates is lost. On the other hand, efficient microbial transformations could open the opportunity to satisfy consumers’ demand of healthy and sustainable foods, while reducing food loss. Considering the AAB role for valorization of date fruits, both acetic acid and gluconic acid fermentations can be exploited for obtaining vinegars, low acetic and gluconic beverages. Although vinegars from dates are already produced, few evidence are available on low acetic and gluconic beverages, which are a valuable industrial segment considering the interest for non-alcoholic, not sugared and functional beverages. The know-how acquired in vinegar field could drive the rational design of processes aimed at exploiting AAB metabolic potential in performing both acetic and gluconic oxidation.”

Round 2
Reviewer 1 Report
/
Author Response
Thank you very much for your comments.
Reviewer 2 Report
Well done, greatly revised.
Author Response
Thank you very much for your comments.
Reviewer 3 Report
The work has been greatly improved according to instructions and I suggest that it be published in its present form.
Author Response
Thank you very much for your comments.